# Exploring the Attitudes of Health Professionals Providing Care to Patients Undergoing Treatment for Upper Gastrointestinal Cancers to Different Models of Nutrition Care Delivery: A Qualitative Investigation

**DOI:** 10.3390/nu13031020

**Published:** 2021-03-22

**Authors:** Kate Furness, Catherine Huggins, Daniel Croagh, Terry Haines

**Affiliations:** 1Nutrition and Dietetics, Monash Health, Monash Medical Centre, Clayton, VIC 3168, Australia; 2Department of Physiotherapy, School of Primary and Allied Health Care, Faculty of Medicine, Nursing and Health Sciences, Monash University, Frankston, VIC 3199, Australia; terry.haines@monash.edu; 3School of Primary and Allied Health Care, Faculty of Medicine, Nursing and Health Sciences, Monash University, Frankston, VIC 3199, Australia; 4Department of Nutrition, Dietetics and Food, School of Clinical Sciences, Faculty of Medicine, Nursing and Health Sciences, Monash University, Clayton, VIC 3168, Australia; Kate.Huggins@monash.edu; 5Upper Gastrointestinal and Hepatobiliary Surgery, Monash Medical Centre, Clayton, VIC 3168, Australia; daniel.croagh@monashhealth.org; 6Department of Surgery, School of Clinical Sciences, Faculty of Medicine, Nursing and Health Sciences, Monash University, Clayton, VIC 3168, Australia

**Keywords:** health professionals, qualitative, upper gastrointestinal, neoplasms, nutrition

## Abstract

Background: People with upper gastrointestinal cancer are at high risk for malnutrition without universal access to early nutrition interventions. Very little data exist on the attitudes and views of health professionals on providing nutrition care to this patient cohort delivered by electronic health methods. COVID-19 has fast-tracked the adoption of digital health care provision, so it is more important than ever to understand the needs of health professionals in providing health care via these modes. This study aimed to explore the perspectives of health professionals on providing nutrition care to upper gastrointestinal cancer patients by electronic methods to allow the future scaling-up of acceptable delivery methods. Methods: Semi-structured qualitative interviews were conducted face-to-face or by telephone and recorded, de-identified and transcribed. Thematic analysis was facilitated by NVivo Pro 12. Results: Interviews were conducted on 13 health professionals from a range of disciplines across several public and private health institutions. Thematic analysis revealed three main themes: (1) the ideal model, (2) barriers to the ideal model and (3) how to implement and translate the ideal model. Health professionals viewed the provision of nutrition interventions as an essential part of an upper gastrointestinal cancer patient’s treatment with synchronous, telephone-based internal health service models of nutrition care overwhelmingly seen as the most acceptable model of delivery. Mobile application-based delivery methods were deemed too challenging for the current population serviced by these clinicians. Conclusion: The use of novel technology for delivering nutrition care to people receiving treatment for upper gastrointestinal cancers was not widely accepted as the preferred method of delivery by health professionals. There is an opportunity, given the rapid uptake of digital health care delivery, to ensure that the views and attitudes of health professionals are understood and applied to develop acceptable, efficacious and sustainable technologies in our health care systems.

## 1. Introduction

Global standards highlight that people have the right to high-quality health care that meets their health needs [1,2]. Evidence suggests that this incorporates the provision of appropriate, timely and expert nutrition care during a patient’s treatment journey, as it is a fundamental determinant of optimal patient experience and outcomes [3,4]. It is well established that malnutrition, and particularly cancer-related malnutrition, has been shown to negatively impact the response to oncological treatments due to increased toxicity leading to dose reductions and incomplete therapy, increased admissions to and length of stay in hospital, increased complication rates, decreases in quality of life with higher morbidity and lower overall survival [5,6,7,8,9,10]. Providing effective nutrition care services to cancer patients is central to mitigating against the adverse effects of malnutrition.

Early, intensive, weekly to fortnightly nutrition interventions have been shown to improve nutritional status in upper gastrointestinal cancer patients undergoing chemotherapy and radiotherapy [11,12]. A possible survival benefit has also been shown in patients who received an early, intensive telephone-based nutrition intervention, commenced at the time of cancer diagnosis [13]. Investigations of current nutrition practices across Australia reveal that the majority of these patients are not assessed preoperatively by a dietitian, despite the existence of clinical guidelines evidencing the value of this [14,15,16], revealing a major gap in nutrition service provision [17]. This is also seen in other countries around the world, where clinical guidelines for the management of nutrition in oncology patients including nutrition screening are implemented ad hoc and at the discretion of individual health services [18,19]. Optimal Care Pathways in Australia exist that clearly delineate all key members of a multidisciplinary team that should be involved in the management of these patients, including a dietitian [20,21]. In practice, this is not always the case, particularly in the outpatient setting [17]. The benefit of preoperative nutrition interventions has been extensively documented in Enhanced Recovery After Surgery (ERAS) programs around the world, which have a focus on preoperative nutrition optimisation inclusive of immuno-nutrition and carbohydrate loading and early resumption of oral diet post operatively [22,23,24,25]. In practice, the use of these guidelines has been poorly implemented. Suggested barriers include lack of awareness of nutrition issues as a priority area, lack of collaboration between dietitians and surgeons, lack of resources and more importantly, the challenge these guidelines pose to the traditional views of surgical practice [9,20].

To overcome these barriers to providing early and intensive nutrition care, we are investigating the effectiveness of different service delivery models, via a synchronous telephone-based approach or an asynchronous mobile technology approach compared with usual care. It is unclear whether these service delivery models are acceptable to health professionals working in the team caring for these patients.

This study aimed to explore health professionals’ perspectives of these two models of service delivery (synchronous telephone approaches and asynchronous mHealth) on the timing, comprehensiveness of communications with the patient and the multidisciplinary team, and what barriers they saw to implementation of these novel nutrition care service models. The perspectives were drawn from health professionals whose upper gastrointestinal cancer patients had been exposed to an early and intensive nutrition intervention delivered using a synchronous, telephone or an asynchronous, mHealth approach.

## 2. Materials and Methods

### 2.1. Methods

This study is part of a three-arm randomised controlled trial testing two home-based electronic health (eHealth) nutrition service delivery models, using either telephone or a mobile application to provide early, intensive, individually tailored nutrition interventions to patients with upper gastrointestinal (UGI) cancers (oesophageal, gastric and pancreatic) as soon as practicable after diagnosis, to determine improvements in quality of life [26].

Design: Semi-structured interviews were conducted either face-to-face or by telephone and recorded with health professionals who were caring for patients with UGI cancers. The Standards for Reporting Qualitative Research (SRQR) was used in the reporting of this study [27].

Eligibility: Participants were eligible to participate if they had awareness of the overarching intervention study and were involved in providing care to patients with UGI cancer.

Setting: The health professionals were employed in one or more of four health services including two tertiary public hospitals and two private hospitals in Melbourne, Australia. The interviews were conducted between March and July 2019.

Participants: Purposive sampling was used to select a group of clinicians that had specific, relevant, and varied insight from a range of health professional backgrounds. The targeted clinicians included luminal upper gastrointestinal and hepatobiliary surgeons (upper gastrointestinal surgeons), medical and radiation oncologists, gastroenterologists (diagnostic endoscopy), cancer care nurses and dietitians.

Trial Registration: Australian and New Zealand Clinical Trial Registry, 27 January 2017 (ACTRN12617000152325).

### 2.2. Procedures

Method of approach: All health professionals were contacted using a scripted email inviting them to participate in the interviews. A follow-up email was sent to outstanding participants over the following two weeks to prompt their response. All participants signed a Participant Information and Consent Form.

Measurements: An interview guide (Table 1) was developed by researchers (CEH, KF). The one-on-one recorded semi-structured interviews were conducted by a single researcher (CEH). The original version of the interview guide was piloted with a health professional who provided feedback to allow further refining of the delivery of the interview and questions. Researchers (CEH and KF) used this feedback and their own reflections of the pilot interview to enhance the interview guide. A scripted introduction to the interview was added to formalise the interviews. Researchers (KF and CEH) met after the fifth interview to discuss progress, content and any changes that might be required to the interview questions/process. This review resulted in more frequent use of prompts to assist with further in-depth exploration of issues. Immediately at the conclusion of each interview, the interviewer (CEH) made reflective field notes.

Researcher Positioning: The researcher (KF) is a senior clinical dietitian in general, upper gastrointestinal and hepatobiliary surgery at one of the tertiary health services in this study. The delivery of the nutrition intervention in the concurrent intervention study was also conducted by this researcher (KF).

Data Preparation: Recordings of the interviews were transcribed verbatim by the researcher (KF) and de-identified for analysis to become intimately acquainted with the dataset and begin preliminary ruminations that would assist with more structured coding and analysis. Transcriptions were sent to participants for member checking. Researchers directly involved with interviewing (CEH) and analysis of the data (KF) both completed face sheets to reflect on the interviews and to record reactions to events occurring throughout the interviews. Recordings, transcribed interviews, and face sheets were all stored in a secure cloud-based repository and erased from the researcher’s recording device.

### 2.3. Analysis

Thematic analysis was chosen as the framework to assist with coding and constructing themes from the interview data to accurately reflect the attitudes and views of nutrition and nutrition care delivery methods of health professionals managing patients with upper gastrointestinal cancers [28,29,30,31].

The verbatim transcribed interviews and face sheets enabled data collection and initial analysis to occur simultaneously and inductively, allowing themes to be constructed and incorporated into the interview process as they arose.

The initial set of themes became the basis for the data analysis framework. This was continually elaborated on as each interview was initially coded for major themes. Themes and subthemes were provided with a label, detailed definitions, and an example quote that demonstrated the theme. When 13 interviews had been completed, KF formally hand analysed each interview. NVivo Pro 12 was then used to sort and analyse the data. This process allowed themes to be re-examined carefully to decide whether further information was required to fill any gaps in the data that had been so far collected. Researchers ((KF), (CEH) and (TH)) discussed the organisation of themes and sub themes until a consensus was reached.

Information power was used as a tool to assist with sample size determination; described by Malterud [32], it indicates that lower numbers of participants are required when greater relevant information is held by the sample.

## 3. Results

### 3.1. Participant Sample

The original identified sample of participants was 33. A total of 13 interviews were completed (38% response rate). There were 18 health professionals that did not respond to invitations to participate and, therefore, were considered non-responders (Figure 1).

Demographics of the interview sample are shown in Table 2.

Participants were predominantly female (69%), surgeons or dietitians (76%) with an age above 40 years (77%). They were most likely to work in the public health system (85%) and have over 20 years’ experience (62%). None of the health professionals outside of dietitians had any formal training or education in nutrition.

### 3.2. Thematic Analysis

Coded data were divided into categories to describe the three main themes:(1)Universal Access to Nutrition Care (Figure 2);(2)Barriers to the Ideal Model (Figure 3);(3)How to Implement and Translate the Ideal Model (Figure 4).

### 3.3. Universal Access to Nutrition Care for People with Upper Gastrointestinal Cancers

Clearly expressed across all health professional backgrounds represented in our sample was the view that provision of nutrition interventions should be universal across the continuum of care, whether the patient was receiving curative or palliative treatments for their UGI cancer. It was clear that the aggressiveness of nutrition interventions should be tailored to align with patient treatment goals.


*#12 “No, I think, no I think it’s never futile, I think that people should be nourished unless it’s truly end of life issues. Then I think there is a role for caring for nutrition”*


### 3.4. The Duration of Nutrition Intervention Covers the Patient Journey

Ensuring patients are adequately nourished was viewed as a core component of patient care, yet many did not think the current nutrition service provision was adequate. Participants viewed that all patients should see a dietitian in an appropriate timeframe, immediately post UGI cancer diagnosis. There were conflicting views on how and when nutrition interventions should be accessed, with some believing that it should be available to access at all times and the patient self-directed a request when it was required to allow individual flexibility; others believed it should be delivered as a blanket service throughout the treatment journey.


*#11 “you know it should be 6 months or 3 months post completion of their treatment. I think that 18 weeks doesn’t actually cover a lot of our patients’ complete treatment”*


### 3.5. Perceived Effectiveness and Trust: The Evidence

Many of the health professionals believed they had good knowledge and understanding of the effectiveness of delivering high quality nutrition interventions to patients with UGI cancers, improving their: nutrition status, ability to have treatment, preparedness for surgery, post-operative outcomes, quality of life and survival. No participant directly cited any literature studies to support these claims. The link between malnutrition and poor health outcomes is well established; it is an assumption of the participants that early and intensive nutrition interventions could bridge the gap between malnutrition and poorer health outcomes.


*#9 “There is very good evidence that nutrition impacts on, directly on survival, impacts the ability to have treatment, improves quality of life. I think there’s very good evidence”*


### 3.6. Anticipated Acceptability and Trust: The Team

When the participants were asked to reflect on the two service models under investigation (synchronous telephone and asynchronous mHealth), they overwhelmingly considered the synchronous, telephone model of nutrition care the most acceptable method given its ease of use, home-based nature, and easy rapport and relationship building with the health professional. They did, however, feel that technology could be better utilised to enhance the synchronous delivery of service by including teleconferencing. The participants speculated that using mobile applications asynchronously was too challenging for the older age demographic of current UGI cancer patients, unless its use was driven by individual choice. Issues highlighted here included unfamiliarity with technology, lack of owning technology and the additional burden of learning new technology, but participants acknowledged that with passing time, it would likely become a more acceptable model.


*#12 “Maybe there is a generational issue, maybe older patients would rather a face-to-face or telephone conversation than using a device or app”*


Communication between health professionals during a patient’s treatment period was considered critical to provision of high-quality care. Health professionals wanted to be able to see what nutrition interventions were delivered and by whom. An internal, hospital-based nutrition care service was the preferred delivery route due to co-location of health professionals in one health service. A small number of health professionals believed that an external, centralised model could enhance access to dietitians and reduce budgetary constraints of multiple internal models. The underlying concerns for an external model included: increasing the fragmentation of an already complex health care system, challenges with access to clinicians to discuss concerns given a lack of established trusting relationships, and concern regarding lack of expertise of the dietitian providing this intervention. Risk of communication breakdown and a distrust of other or unknown health professionals was identified as key.

Many patients with UGI cancer access many different health professionals over the span of their cancer journey, and many of these are not provided within one health service. Our participants viewed providing integrated nutrition care, from internal and external health care sites across metropolitan, regional and rural health services as important to ensure continuity of nutrition care, yet noted challenges with dietetic speciality expertise and communication channels back and forth as major barriers to this. Both methods of nutrition service delivery tested in the overarching trial could act as a bridge to perceived lack of dietetic expertise in rural/regional areas. Our participants valued the access they had to specialty dietitians within their health service and proposed that they could also support community, regional and rural dietitians through training and/or access to their expertise as subspecialists.


*#6 “So, better partnerships with our regional hospitals would be really important. It’s around resource and kind of service delivery, understanding our patient journey and where our patients are being seen, and going”*


While team dynamics broadly underpinned the provision of optimal overarching health care to patients inclusive of nutrition interventions, trust was described as encompassing established respectful relationships built over time, good communication with a common goal and purpose clearly identified.


*#1 “I suppose it’s about those relationships, it’s about the staff here, the premise is that the medical staff actually know the dietitian. I’m going to say respect but feel comfortable and confident in their clinical skills and the information they give”*


### 3.7. Resource Scarcity

Whilst our health professionals hold a view that universal provision of nutrition interventions should be provided across the continuum of care, this is not translated into everyday practice. They perceived that the major barriers to provision of nutrition care were resource constraints—not only financial barriers but time and accountability barriers, where other aspects of treatment are prioritised at the expense of nutrition provision.


*#9 “Well I think that it’s well recognised that nutrition is incredibly important, however where the resources are so ahh poor, you know, both in umm medical and allied that we really don’t optimise nutrition and there is lots of data around malnutrition in patient populations obviously, so it’s a real issue for us”*


### 3.8. Who Is Accountable?

These resourcing constraints also lead to medical clinicians not being able to or wanting to take responsibility for their patient’s nutrition and/or weight loss. Overarching organisation systems of accountability management including adequate and encompassing communication channels via the electronic medical record were described as lacking, so that health professionals and teams were prevented from taking full responsibility.


*#11 “It’s very, I think it’s very hard for one person to take ownership and claim ‘I need to be responsible for their weight loss’”*


### 3.9. Trust: Us and Them

Underscoring accountability were the challenges in how to manage a patient’s nutrition with an undercurrent of clinicians, generally dietitians versus medical clinicians, holding very different opinions, often opposing, that led to frustrations, tensions, and dysfunctional team dynamics despite our participants all agreeing that universal nutrition care was essential.


*#1 “Their view (medical staff) is, they’re fine because they’re, you know they’ve got a BMI of 23 or whatever but in actual fact they may not be fine. We forget that in, they are living off soup in the short term, but they’re eating, so, it is about changing their minds as well”*


These attitudinal differences were at play when medical clinicians prioritised and managed a patient’s nutrition.


*#3 “I think you’ll find there is some traditionalists who ‘I’ve done this the same way always, my patients do very well, I fit within the ANZGOSA death and complications guidelines’”*


### 3.10. Battle of the Norms

Battle of the norms was seen as a barrier to implementation, where tension exists between the view that unintentional weight loss and subsequent malnutrition were seen as a normal part of the UGI cancer journey despite a view that the provision of nutrition intervention should be universal.


*#6 “I think there is an apathy towards weight change in our oncology patients, especially in our UGI in that there is an acceptance by our medical teams and nursing staff that all patients will lose weight on their journey”*


### 3.11. Lack of Access

In the ideal model, health professionals expressed the desire for improved, timely access to nutrition services. It was highlighted as a major barrier to the provision of nutrition interventions to their patients. There were a number of identified concerns relating to access including: lack of early (pre-treatment) access to dietitians, lack of dietitian availability in speciality outpatient clinics both due to resource constraints and allocation, the fragmented nature of outpatient clinics where health professionals are not co-located, time lag to access dietetic clinics, and perceived lack of specialty dietitians managing these clinics.


*#9 “So, we’ve now got a 6–8 week waiting time for a new patient and in that time, if you’ve got oesophageal cancer, you’re going to lose a lot of weight and even block your oesophagus off. Whereas if you started treatment earlier, so it’s not really acceptable but that’s what we’ve got”*



*#11 “*
*It’s just that pre-treatment part, the from the point of diagnosis to treatment is the part where we have a big, big issue and a big downfall”*


### 3.12. Governance as Driver

Health professionals discussed a range of systematic national, state, and organisational mechanisms to enhance accountability and improve the quality of patient care. These included: standard operating procedures delivered by surgical governing bodies, state-wide optimal care pathways, and organisational nutrition and hydration committees who were responsible for operationalising and monitoring key performance indicators related to nutrition.


*#3 “Once we’ve got the evidence and the information out there, that’s when a group, a governing body like ANZGOSA which is appointed by the College of Surgeons to run upper GI surgery in this country can send out a draft position paper to all upper GI surgeons in the country”*


### 3.13. Trust: Who Is Accountable?

Accountability is underpinned by governance structures, assisting to find solutions to providing nutrition care to UGI patients, with all multidisciplinary team members playing a role in sharing the responsibility for managing nutrition throughout a patient’s journey.


*#6 “Well how do we get our medical buy in?’ because that’s more important I think”*


### 3.14. Systems Change

To embed governance and accountability, the overarching complex health care system and the current patterns of working were viewed as needing a major refocus. Changes included: appropriate resource allocation, dietetic involvement in multidisciplinary team meetings to promote collaboration and joint problem solving, enhanced models of communication, patient-centred, culturally appropriate interventions and service delivery, multidisciplinary outpatient services, monitoring, evaluating and reporting nutrition outcomes and embedding, enhancing and keeping pace with technological advancements.


*#2 “If we could get a model and we are so far off this but would replicate something like what the breast patients get. And that’s the one stop shop. The come in and they see their surgeon, they see their oncologist, they see their dietitian, they see their nurse…”*


### 3.15. Develop and Expand the Workforce

Within health systems change was the view that developing and expanding the workforce of nutrition specialists was a way to move forward. This would incorporate senior specialist dietitians in tertiary centres sharing knowledge with other dietitians, including community based-dietitians, junior staff, and students.


*#9 “The other thing is sort of community based dietitians, but they would need a competency in this area. Umm, and I know for psychology umm, the MPCCC is doing a project looking at upskilling community based psychologists knowing that we’re never going to have enough in the hospital system, then if you had community based, there’s enough people with cancer in the community that you could have practices that, that you know, specialise in this”.*


Participants also recognised that in a resource-scarce environment, expanding the workforce to enable other health care professionals to be able to deliver fit for purpose nutrition education and/or interventions, such as allied health assistants (AHAs) and/or cancer care coordinators/nurses, was also viewed as a consideration.


*#6 “We need the right person to be delivering the right task at the right time and that doesn’t necessarily need to be a specialist. I think there is a care partnership that would be possible”*


### 3.16. Teach and They Will Come

Dietitians expressed their concerns with a lack of knowledge relating to the importance of nutrition as a contributing factor to how nutrition is prioritised by medical and nursing staff.

It was felt that there was a need for innovative ways to provide nutrition education to different health disciplines. Many believed that embedding nutrition education formally in medical/nursing training through universities and/or healthcare organisations would be beneficial to enhancing the provision of nutrition to this patient population. Yet, a medical clinician believed that informal nutrition training throughout medical and sub-specialty training should be the focus rather than being the domain of universities to teach.


*#6 “So I think that having nutrition taught in a medical degree is really important and should be one of the core subjects for them”*


### 3.17. Rapid Translation

As nutrition research expands our field of knowledge on the management of this patient cohort, there is a significant time lag between research evidence and translating that into clinical practice. Translating evidence in a time-efficient manner to improve health service delivery was important to clinicians across the board. Practical solutions to this included ensuring evidence is presented at surgical/oncology conferences with the aim of reaching large audiences and improving knowledge of best practice. There was a gap in broadening this to incorporate other formats of research and information dissemination to engage the largest number of stakeholders possible.


*#3 “The upper GI/oncological/radiotherapy/surgical community, we present, we produce papers, present at meetings, present at our craft group conferences. The only way to get surgeons to change is to get up there and talk about it”*


## 4. Discussion

This study is the first to explore the attitudes and views on the delivery of nutrition interventions to people with upper gastrointestinal cancer using eHealth from a wide range of health professionals. We were able to gain a broad understanding of clinicians’ views on nutrition care by drawing on participants’ concurrent expertise with a novel nutrition care intervention. Whilst nutrition was viewed as an important part of cancer care, there were concerns with barriers to provision of nutrition interventions. Acceptance of novel nutrition delivery modes is complex. When asked to ideate about the optimal nutrition service, clinicians wanted early and accessible care throughout the treatment period, trust, good communication and cohesion within teams, and integration across health services. Conversely, in practice, currently there is an acceptance, particularly from our medical specialists, that patients’ weight loss is part of the disease progression and/or treatment side effect. We found that there was a lack of accountability with regard to who should be addressing weight loss as a treatment goal.

Trust was a major theme that permeated throughout the analysis. Interprofessional health care teams are defined as those with professional diversity [33]. Effective, high performing teams have been shown to enhance the quality and safety of health care provision, which ultimately leads to improved outcomes for patients [34,35,36,37]. Conflict that leads to friction within teams has been described as affective conflict [37], where individuals with strong professional identity are driven by the priorities of their own profession [38]. This can lead to low levels of collaboration through decreased receptiveness to differing perspectives and views. Some proposed ways to enhance trust within teams include: high levels of role clarity, identification of shared goals and values using a top down approach, transparent processes, thorough examination of competing ideas, breaking down of hierarchical culture and empowerment of trust, where trust facilitates performance [36,39].

All clinicians viewed the provision of universal nutrition care to all patients across the continuum of care as one the most important facets of an ideal model of care, and this included patients that were for curative or palliative intent treatment. This aligns with the universal model of healthcare that Australia maintains at the centre of its publicly funded Medicare system [40]. When asked how care should be provided, practitioners described ongoing care throughout the patient’s treatment journey. However, there is a need to ensure that the costs associated with providing aggressive health care, including nutrition care, to this population, are justified. It could be argued that providing intensive nutrition interventions in this population is futile care, where futility refers to the provision of interventions that do not produce any significant benefits to the patient [41]. This is unlikely though, as the provision of timely nutrition support has been shown to improve weight stabilisation, symptom management, quality of life and survival [42,43].

To deliver this universal nutrition care, health professionals viewed internal health service, synchronous delivery methods encompassing face-to-face, telephone and/or teleconferencing as the acceptable model, whereas they held the view that the asynchronous mobile application delivery methods would be challenging for the patient to operate. Use of normalisation process theory (NPT), which specifically focuses on the action of the implementation process rather than beliefs and attitudes, has been used successfully to change health professionals’ behaviour [44]. The key components that are required include modifying and reinforcing peer group norms and expectations, emphasizing the complexity of social relationships and interactions in changing behaviours of a closely aligned cohort of health professionals [44]. This aligns to the Diffusions of Innovation Theory where an opinion leader acts to influence the behaviour of adopters to perform the new and innovative behaviour [45]. The head of medical and surgical units and the overarching healthcare organisation could act as these change drivers to adopt eHealth models of health care delivery.

The high prevalence of weight loss and malnutrition identified in UGI cancer patients was seen by many as a normal part of the disease and treatment course. This was an element of the complex interplay of the differing views of accountability which also encompassed attitudes of clinicians to prioritise and manage weight changes. Early nutrition interventions are strongly recommended for patients receiving chemotherapy, radiation therapy and/or surgery for upper gastrointestinal cancers with individualised, weekly to fortnightly reviews [10,13,14,15]. Early and intensive nutrition support prior to surgery is endorsed by much of the literature in the prehabilitation and ERAS space, yet there was no mention of these as a model of surgical care [22,23,24,25,26,27,28]. There is a real opportunity for nutrition to become a monitored management strategy and assist with addressing compliance in treating malnutrition through broadening the scope of cancer registries to incorporate nutrition status as part of the core set of indicators to improve quality of care and survival for patients.

Health professionals were concerned with their patient’s ability to access nutrition care (and cancer care more broadly) in a timely manner, with time delays increasing with growing numbers of patients attempting to access these services and these services being stretched beyond capacity. eHealth has the potential to bridge this access gap. This is important because accessing nutrition care earlier can positively impact a patient’s illness trajectory [10,11]. Resourcing decisions are often based on historical allocations, and this needs to be explored in detail given the unmet needs of this patient population, which has evolved significantly over time [46]. The novel delivery models used in our intervention study provide an answer to many of the issues associated with accessing dietetic services. Engaging our health professionals will require robust evidence of the usefulness and ease of use of these delivery models before there will be widespread support for their use, as these have been identified as the two key components to the acceptance of eHealth [47].

Medical and nursing staff’s nutrition knowledge was a focus for many discussions with dietitians and viewed as a barrier to attitudinal change and ultimately, accountability. A mixed method study by Awad [48] surveyed 63 surgeons and 25 dietitians and found that surgeons’ knowledge of nutrition support principles was poor despite varying degrees of seniority, yet they were regularly making decisions related to the nutrition support of their patients. A six-country comparison of undergraduate medical curricula including Australia found that there are no specific nutrition competencies directed by the Australian Medical Council (AMC) [49], and although a framework to incorporate this does exist, it is not mandated, presenting a significant barrier to uptake amongst universities [50].

We acknowledge the limitations in our study. The participants in the study may have been more likely to report favourable opinions about nutrition interventions because of their knowledge of the RCT being conducted within their health service. For example, health professionals told us that universal nutrition provision was the ideal model, but they alluded to other health professionals having different perspectives as evidenced by the lack of referral to dietitians and accountability of their patient’s nutrition status. We also had a limited sample size. All participants were from a small number of Victorian health services and speciality areas, which limits the generalisability of our results to similar health services.

This study has identified several important areas for future research. Themes related to trust permeated throughout the different aspects of our analysis, indicating that better dissemination of existing research into the effectiveness and cost effectiveness of nutrition interventions’ delivery via eHealth approaches may be required before some team members become convinced of their value. We need to challenge the assumption held by the current group of health professionals that asynchronous mobile application-based nutrition delivery models are not appropriate for their patients. The health professionals’ investment in learning how these technologies could deliver timely and accessible nutrition care may lead to change in their attitudes. Co-design and testing of nutrition models with medical, nutrition health professionals and patients are warranted.

## 5. Conclusions

This study found that health professionals involved with treating patients with upper gastrointestinal cancers viewed the provision of nutrition interventions as an essential part of their overall cancer treatment. Their suggestion on how these interventions should be delivered to patients in a timely and accessible manner mimicked the current predominate delivery mode which is largely face-to-face. Synchronous, telephone-based services were viewed as a potential strategy, with the preference for an internal health service model of nutrition care. Trust within and outside of teams was viewed as important to facilitate cohesion and integration across health services. Governance structures were regarded as a way to manage many of the overarching barriers associated with implementing a successful model of nutrition care. This study highlights the need to gain deeper understanding of the attitudes and behaviours of our health professionals in managing nutrition concerns such as weight loss of this high-risk patient group. The use of novel technology for delivering nutrition care for people receiving treatment for UGI cancers was not widely accepted as the preferred method of delivery by health professionals. There is a need to develop greater acceptance and adoption of these technologies for the provision of health care as this may help in achieving the goal of universal access to nutrition intervention.

## Figures and Tables

**Figure 1 nutrients-13-01020-f001:**
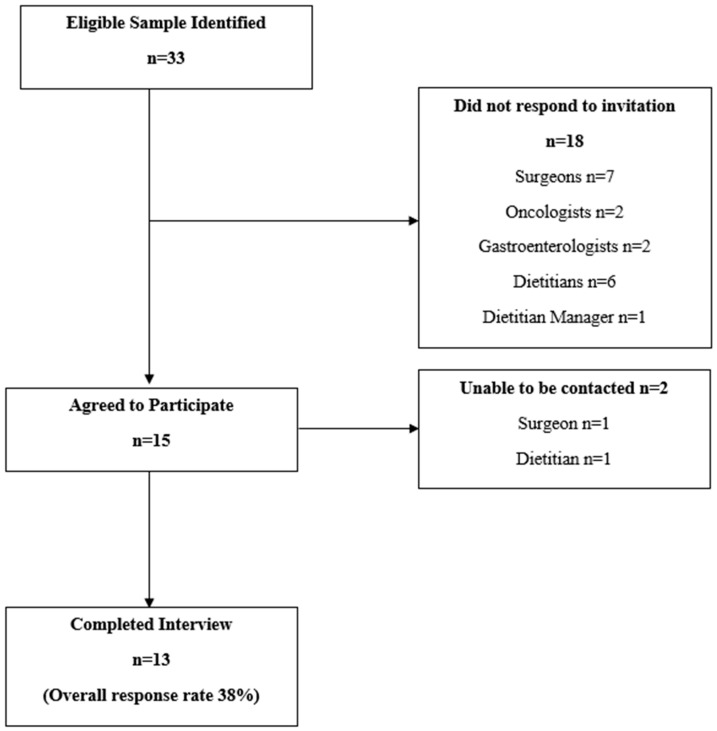
Participant flow.

**Figure 2 nutrients-13-01020-f002:**
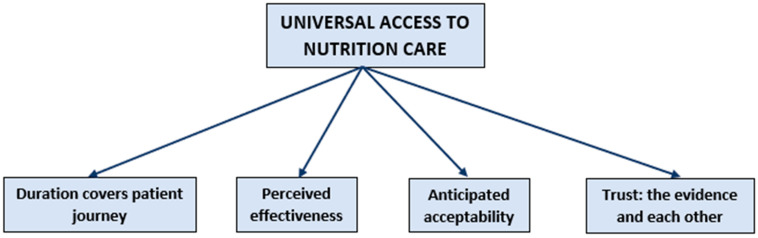
The model of Nutrition Care Idealised.

**Figure 3 nutrients-13-01020-f003:**
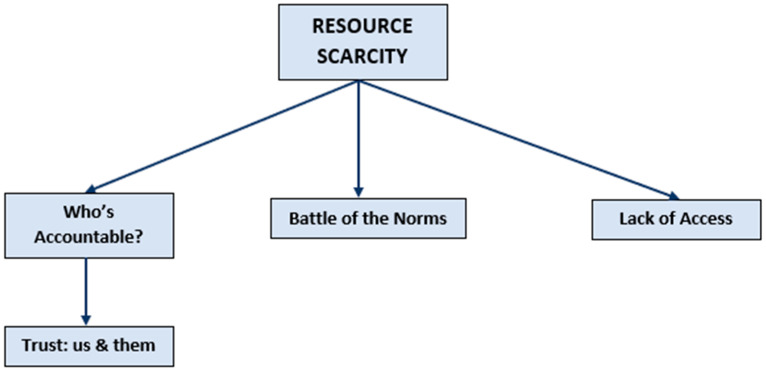
Barriers to the ideal model.

**Figure 4 nutrients-13-01020-f004:**
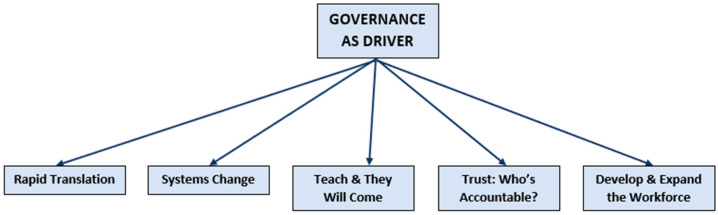
How to implement and translate the ideal model.

**Table 1 nutrients-13-01020-t001:** Interview guide.

Question	Question Logic
Describe to me your role in managing patients with upper GI cancer?	Elicits background information about the health professional role
Cancers of the upper GI tract are often associated with weight change. Tell me about your experiences as a practitioner and managing cancer treatment in patients who are experiencing weight change.	Explores the views, experiences and attitudes of health professionals managing weight change in their patients
Given that upper gastrointestinal cancers have very low survival rates, do you think that there is any benefit to providing nutrition interventions to these patients?	Explores the views of health professionals of the provision of nutrition interventions across the continuum of care
All of your patients who were diagnosed with gastric, oesophageal, or pancreatic cancers were eligible to access early and intensive nutrition support via a telephone or mobile app delivery method. Can you tell me what experience you have had with this research study?	Elicits information about health professional’s knowledge of the intervention study and what experiences they had with it directly
Did you experience any challenges with the study, if so, can you give some examples?	Ascertains any challenges they had with engaging with the intervention study
Tell me about your opinion of the two different delivery methods?	Explores the views of health professionals regarding external, centralised synchronous and asynchronous nutrition care delivery models
If you could design a service delivery model for your patients what would the features be?	Explores the ultimate nutrition care service delivery model
To access this nutrition service delivery model, describe the referral process that would work for you?	Ascertains information about the best nutrition referral method for health professionals
If you could have one thing at your disposal to address the nutrition concerns of your patients, what would it be?	Elicits information about the most important factor to implementing the best nutrition care model
Is there any final remarks or comments you would like to make?	Open commentary from health professionals

**Table 2 nutrients-13-01020-t002:** Demographics of participant sample.

Demographics		
Profession	Total	Percent
Surgeon	5	38%
Oncologist	1	9%
Nurse	2	15%
Dietitians	5	38%
Gender		
Female	9	69%
Male	4	31%
Age		
30–40	3	23%
40–50	6	46%
50–60	4	31%
Health Service		
Public	7	54%
Private	2	15%
Both	4	31%
Experience (years)		
5–10	3	23%
10–20	2	15%
20–30	7	54%
30+	1	8%

## Data Availability

The data that support the findings of this study are available from the corresponding author upon reasonable request.

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
