# Peer review of "Exploring the Attitudes of Health Professionals Providing Care to Patients Undergoing Treatment for Upper Gastrointestinal Cancers to Different Models of Nutrition Care Delivery: A Qualitative Investigation"

_nutrients, 2021, doi:10.3390/nu13031020_

Round 1

Reviewer 1 Report

Thank you for the opportunity to review this paper which looked to explore the perspectives of health professionals in providing nutrition care to upper gastrointestinal cancer patients by electronic methods to allow the future scaling-up of acceptable delivery methods. This is a well written and interesting paper, I have a few small comments below. It also appears that the main message in the conclusion – that HCPs don’t support mhealth is not one of the main themes in what has been reported in the paper

Abstract

  • In the conclusion the abbreviation UGI is used for first time

Introduction

  • The first sentence indicates a large source of support yet only one reference is provided. Also is this not an agreed standard globally?
  • Be good to expand the relevance of nutrition, particularly specific to upper GI cancers

Methods

  • Page 3, line 114 and line 121 and lines 131-135– change in font size
  • Table 1 – indicate what TEND stands for

Results

  • Page 6, line 166 – it appears that surgeons and dietitians combined make up 38% rather than 38% each therefore 76%
  • Page 6, lines 180-181 – repetition of word clear/clearly
  • Page 6, line 189 – sentence doesn’t make sense as currently written, think ‘as’ should be ‘was’
  • Page 6 – duration is not included as a heading while all other themes in Figure 2 are included as headings
  • Page 8, line 279 – the subtheme ‘trust, them and use’ doesn’t seem to really reflect the information provided which seems to detail differing views on what is right

Discussion

  • Page 13, line 466 – conduction is a typo, should be conducted

Conclusion

  • Conclusion regarding lack of acceptance of mhealth is very strongly worded given more prevalent themes and the limitations of the study e.g. sample size

Reviewer 2 Report

The manuscript deals with a very important topic, dietary care and nutrition of patients diagnosed with cancer of the gastrointestinal tract.

line 58-59. Do the authors have data on Investigations of current nutrition practices across USA or Europe countries?

In the introduction, the authors should describe whether in Australia, in hospitals treating oncological patients, there are nutritional teams consisting of, for example, a surgeon, gastroenterologist, dietitian and psychologist.

Figure 1 - it is a pity that such a small percentage of health professionals invited to the study agreed to participate in the study. It would be important to study a larger group of doctors of different specialties who are involved in the treatment of patients with gastrointestinal cancer.

Table 2 - Nurses also took part in the study, whether they also had training in nutrition. This is important information because it may affect the answers provided.

line 383 - nutrition was viewed as an important part of cancer care - there is no presentation of the opinion of specialists depending on the specialty. Were there differences in the approach between dieticians and surgeons? which would give a picture of the approach to the patient.

Line 388- Conversely, in practice, currently there is an acceptance that patients’ weight loss is part of the disease progression and/or treatment side effect - did the nutritionists think so in the study?
